# Associations Between Stress Level, Environment, and Emotional and Behavioral Characteristics in Service Sector Employees

**DOI:** 10.3390/ijerph22091390

**Published:** 2025-09-05

**Authors:** Sylvie Rousset, Carole Brun, Gil Boudet, Philippe Lacomme, Frédéric Dutheil

**Affiliations:** 1UNH, UMR1019, INRAE, University Clermont Auvergne, 63000 Clermont-Ferrand, France; 2LIMOS UMR CNRS 6158, University Clermont Auvergne, 63000 Clermont-Ferrand, France; placomme@isima.fr; 3Physiological & Psychosocial Stress, LaPSCo, CNRS, University Clermont Auvergne, 63000 Clermont-Ferrand, France; gil.boudet@uca.fr (G.B.); frederic.dutheil@uca.fr (F.D.); 4Prevent & Occupational Medicine, University Hospital Clermont-Ferrand, 63000 Clermont-Ferrand, France

**Keywords:** perceived stress, food intake, physical activity, emotion, tertiary sector employee, overweight, digital recordings

## Abstract

**Background**: The prevalence of stress-related health issues is becoming increasingly significant. This study aimed to examine the relationships between work stress, home stress, overall stress, and individual behavioral and perceptual characteristics among middle-aged employees in the service sector. **Methods**: Physical activity, diet, and perceptions were assessed using the WellBeNet application (2.10.2, INRAE, Clermont-Ferrand, France) while perceived stress levels were evaluated through an online questionnaire during a one-week period. The associations between stress levels and individual and behavioral characteristics were examined using multiple linear regressions and analyses of variance. **Results**: General stress was significantly influenced by both work and home stress. Home stress was positively correlated with the perception of one’s silhouette in red, the increasing consumption of dairy products, and the decreasing consumption of vegetables. Work stress was inversely correlated with age and positively correlated with body shape. **Conclusions**: Our study identified various context markers of stress—including age, body shape, food intake, and color of the silhouette. These markers could be used in subsequent intervention studies to demonstrate causal links.

## 1. Introduction

In today’s rapidly changing and complex society, a growing number of individuals are grappling with stress-related issues [1]. Stress can have a profound impact on one’s health and can lead to physiological disorders. Therefore, by altering the functioning of the autonomic nervous system, stress disrupts the sympatho–vagal balance, heart rate, and its variability, often with lasting effects [2]. The reduction in heart rate variability is associated with cardiac events in adults; it is a risk factor for high blood pressure or stroke [2]. Stress can give rise to psychological distress, anxiety, and depression, which in turn impact one’s behavior. In the general population, psychosocial stress is linked to decreased physical activity, particularly during leisure time, and increased screen time [3]. Stress is known to heighten impulsiveness while reducing emotional, visceral, and behavioral control [4]. Prolonged stress amplifies the consumption of highly pleasurable sweet and fatty foods [5] while diminishing the intake of fruits and vegetables [6]. These behaviors serve as coping mechanisms. If this coping strategy persists, it may contribute to weight gain, obesity, and metabolic disorders [7].

Perception of stress is influenced by various factors, both individual and environmental. A study revealed that the prevalence of moderate stress was lowest among individuals aged 60–69 years, while the highest prevalence was observed in the 40–44-year age group [8]. Women between the ages of 30 and 34 reported a higher frequency of moderate stress compared to men within the same age range. Additionally, the prevalence of high stress was more common among women than men [8]. A correlation was found between stress and obesity [6]. Both work and family environments can present challenges for individuals. In the workplace, stress can arise from various sources, including career development, relationships with colleagues, and work organization [9]. Managerial and administrative workers tended to express greater satisfaction with their career positions compared to manual workers. Some professions are more stressful than others, such as those requiring the most physical or psychological efforts or those providing the least satisfaction [10]. Workers with higher education had a tendency for lower levels of perceived stress. Hard manual workers or persons with physical work had a higher risk of a high perceived level of stress compared to managerial and administrative workers in the tertiary sector [11]. The latter were more satisfied with their achieved career position than manual workers. However, employees in the tertiary sector, who typically work in an office setting, face an increased risk of experiencing high stress levels if they spend more than an additional 3 h per day in a sedentary position outside of work, which can multiply their stress levels by 3 or 4.

This cross-sectional study explored the individual characteristics, behaviors, and emotions linked to stress among employees in the tertiary sector. The purpose of the study is to first examine the relationships between general stress and stress at work and at home, then between stress and behavioral and perceptual variables, depending on the context. The study was an observational survey that was designed based on data collected remotely thanks to the participants’ smartphones and computers. The cross-sectional study was chosen because it is suited to the study of covariation relationships between stress and the large number of new variables collected using our original device, the WellBeNet application (Version 2.10.2, INRAE, Clermont-Ferrand, France). Moreover, cross-sectional studies are suitable for mature populations like ours that have reached equilibrium and are not vulnerable to transitory biases [12]. In this case, one time point can be representative of a stable state and provides relevant information on the relationships between stress, behavior, and perceptions.

## 2. Materials and Methods

### 2.1. Study Design

This study aimed to objectively assess physical activity and sedentary behaviors and subjectively assess self-reported stress levels, emotions, and food choices in a voluntary sample of tertiary employees aged 40–60 years (Figure 1). Data collection took place over one week during a working period based on the availability of participants who used their smartphones and computers remotely.

### 2.2. Participants and Selection Process

Participants were recruited through social networks and mailing lists of the Auvergne Rhone Alpes INRAE Center and the University of Clermont Auvergne. Eighty candidates responded to the advertisement, but fourteen did not have a smartphone compatible with the WellBeNet application. Fifteen withdrew before the start of the study, because they were ill with COVID-19, had a problem with their phone, or did not have enough time to take part in the study (Figure 1). The study was conducted in accordance with the Declaration of Helsinki and approved by the Committee for the Protection of Human Subjects (Sud-Est VI, reference 2020/CE 06). Prior to completing the survey, participants provided informed consent online. Recruitment of participants began on 14 March 2022 and ended on 3 June 2022.

### 2.3. Main Outcome

Perceived Stress: To assess stress levels at work, at home, and in general, three visual analogue scales were used [13]. These scales consisted of horizontal continuous lines of 100 mm, ranging from “very low perceived stress” on the left to “very high perceived stress” on the right, without graduation. Participants were instructed to report their stress levels at the same time each evening throughout the week.

### 2.4. Secondary Outcomes

Body Mass Index (BMI)**:** BMI (kg/m^2^) was calculated using self-reported height and weight provided by the participants. Participants are considered overweight when their BMI is between 25 and 30 kg/m^2^ and obese when their BMI is above 30.

Physical Activity: Participants were instructed to download the WellBeNet app (version 2.10.2, INRAE, Clermont-Ferrand, France) from the Play Store and provide their age and sex to the researcher. They then input their height and weight into the app and were directed to use the eMouve, NutriQuantic, and EmoSens components of the WellBeNet application (Version 2.10.2, INRAE, Clermont-Ferrand, France) for seven consecutive days. To collect accelerometry data during waking hours, participants were instructed to wear their smartphones in their pants pocket. Time spent in four activity categories (immobility, light-intensity activity, moderate-intensity activity, and vigorous-intensity activity) was calculated as a percentage of total recording time. Notably, eMouve provided accurate estimations of time spent in these categories, with minimal errors for both normal-weight and overweight subjects [14,15]. Guidoux et al. (2014) explain how the physical activity/sedentary behavior assessment algorithm was designed [16]. The absolute errors in estimating the time spent in the four activity categories (sedentary, light-, moderate-, and vigorous-intensity activity) are less than 5% compared to indirect calorimetry in normal-weight or overweight adults.

Eating behavior: NutriQuantic was used to track the number of food portions consumed during the same period, regardless of the food category. A portion estimation guide was provided to each participant, including examples of portion sizes for different food categories (Table 1). The number of portions in each category was counted, and the proportion of each category was calculated by dividing the number of portions by the total number of food portions. A nutritional score was assigned to each of the 11 food categories (alcohol, hot drink, starchy product, fruit, nut, dairy product, legume, vegetable, fatty–salty–sugary product, snack, and meat–fish–egg) based on the number of portions consumed, following French and international nutritional guidelines [17]. The assigned score ranged from 0 to 1, and a nutritional balance score for the overall diet was calculated based on confidential calculations using the food categories [18]. A nutritional score of 0 corresponds to a number of servings that is very far from the nutritional recommendations, 0.5 to a number that is slightly far from the recommendations, and 1 to a number of servings in line with the recommendations. The balance score is the sum of the individual scores.

Perceptions: With EmoSens, participants assessed their body shape, physiological state (desire to eat and move), and emotions in the evening [19]. They selected one of nine silhouettes defined by Stunkard et al. (1983) that corresponded to their body shape, and colored it with one of the following hues: orange, yellow, red, gray, or white [20,21,22]. The choice of color was made spontaneously without any information about the meaning of the color. Each hue represented a different emotion or state. The color red is often associated with a high level of excitement (passion, anger, perception of danger, etc.), orange with a lower level of excitement, yellow with joy, white with a neutral emotion, and gray with sadness [23]. Participants rated their emotions using a 10-point scale based on the Geneva Wheel and scored their desire to eat and move on unstructured scales, with scores ranging from 0 to 100 [24]. For each day of recording, we calculated the mean values of positive and negative emotions. Then, we calculated the mean values of positive and negative emotions and of the two desires over the week. The proportions of color choices were calculated over the week.

All data collected by smartphones is sent anonymously to the ActivCollector server and processed anonymously [25].

### 2.5. Statistical Analysis

Normality of variables was tested using the Shapiro–Wilk test, quantitative data are presented as mean and SD, and qualitative data are presented as a number (%). To compare the number of executive vs. non-executive employees in each weight status taken 2 by 2, we carried out chi^2^ tests. A significant *p*-value (<0.05) indicated an association between weight status and professional category. Multiple linear regression models were used to analyze the relationships between general stress, stress at work and home, and individual and behavioral characteristics. Results of linear regressions were expressed as r^2^, and regression coefficients with their 95% confidence intervals were shown. The SAS proc REG with the stepwise option was used (Version 9.4, SAS Institute, Cary, USA). For variables that exhibited a normal distribution, differences between levels of perceived stress (low, medium, and high) were assessed using analysis of variance (GLM procedure of SAS). Non-normally distributed variables were examined using Cochran–Mantel–Haenszel statistics (cmh) for differences in row mean ranks (FREQ procedure of SAS). A significant *p*-value (<0.05) indicated a strong association between stress level and variable value, while a *p*-value between 0.05 and 0.10 indicated a trend. For variables indicating which differences between stress levels were found to be significant, multiple mean comparison tests (LSMEANS) were conducted for normally distributed variables. In the case of non-normally distributed variables, differences between the low and medium stress levels, medium and high stress levels, and low and high stress levels were examined using Cochran–Mantel–Haenszel statistics. Furthermore, effect size (*ES*) was calculated using Hedges’ g equation (Equation(1), [26]): (1)$$ES=\;\frac{mean\left(F1\right)-mean(F2)}{SDpooled}\;$$
where *F*1 and *F*2 represent two stress levels among the three (low, medium and high), and *SDpooled* is the pooled standard deviation.

## 3. Results

### 3.1. Participant Characteristics

A total of 51 participants (41 women and 10 men) were included in the study (Figure 1). The sample had a notably lower representation of men. The average age of the participants was 48 years. Among them, 24 participants were classified as being of normal weight, 15 were categorized as overweight, and 12 were classified as obese (Table 2). All have intellectual office jobs without significant physical activity. Twenty of them are executive and thirty-one are non-executive employees. A one-way analysis of variance does not show differences in age depending on weight status (F = 1.02, *p* = 0.37). Chi-square tests show that the number of executive employees was slightly higher among normal-weight individuals than among overweight individuals (χ^2^ = 2.83, *p* = 0.09) or obese volunteers (χ^2^ = 2.75, *p* = 0.09). There was no difference in the number of executive and non-executive employees among overweight and obese volunteers (χ^2^ = 0.01, *p* = 0.92). However, it is important to note that six of the fifty-one participants did not complete the stress questionnaire (five women and one man).

### 3.2. Perceived Stress

The mean (SD) values of perceived general, work, and home stress were 30.5 (17.5), 31.2 (20.7), and 21.2 (16.6), respectively (Figure 2). The mean values of general and work stress were similar and higher than the mean home stress value. There was significant variability in stress levels around these means, enabling us to categorize participants into three groups of equal size.(2)GeneralStress=7.9+0.6×HomeStress+0.3×WorkStress

The 95% confidence intervals of regression coefficients are [1.1 to 14.8], [0.3 to 0.8], and [0.1 to 0.5] for the intercept and the coefficients associated with stress at home and stress at work (Equation (2)).

### 3.3. Linear Relationships of Correlation Between Perceived Stress and Individual, Behavioral, and Emotional Characteristics

The multiple regression model shows that general stress was explained by two components of stress, home stress (r^2^ = 0.46, *p* < 0.0001) and work stress (r^2^ = 0.13, *p* = 0.0006), with a total explanation of 59% (*p* < 0.0001, Figure 2, Equation (2)). These two components were partly dependent (r^2^ = 0.13, *p* = 0.01) and associated with different variables. In a multiple regression model for explaining home stress, two emotional variables and two variables related to diet were relevant (r^2^ = 0.41, *p* = 0.0002; Equation (3), Figure 3). Choosing the color red instead of orange for one’s silhouette was indicative of high levels of home stress (r^2^ = 0.22, *p* = 0.001). Consuming fewer vegetable portions (r^2^ = 0.09, *p* = 0.02) and a higher percentage of dairy products (r^2^ = 0.07, *p* = 0.04) were also features associated with high levels of home stress.(3)HomeStress=23.9+4.8×dairyprod−6.9×vegetable−10.7×orange+24.9×red

The 95% confidence intervals of regression coefficients are $\left[12.7\;to\;35.3\right],\;\left[-0.1\;to\;9.7\right],\;\left[-12.1\;to-1.7\right]$, $\left[-23.6\;to\;2.3\right]$, and [7.3 to 42.4] for the intercept, the coefficients associated with the percentage of dairy products, the number of vegetable portions, and the orange and red shades used to color the silhouettes (Equation (3)).

In a third regression model, work stress was explained by two complementary individual variables (r^2^ = 0.26, *p* = 0.002; Equation (4)). Being younger and perceiving one’s silhouette as overweight were associated with high levels of work stress (r^2^ = 0.19, *p* = 0.003, r^2^ = 0.07, *p* = 0.06, respectively; Equation (4), Figure 3).(4)WorkStress=91.0−1.7×age+4.8×silhouette

The 95% confidence intervals of regression coefficients are $\left[30.2\;to\;151.9\right],\;\left[-2.8\;to\;-0.6\right]\;and\;\left[-0.3\;to\;9.9\right]$, for the intercept, and the coefficients associated with age and silhouette (Equation (4)).

### 3.4. Effect of Home Stress Level on Individual, Behavioral, and Emotional Characteristics

Analyses of variance revealed significant effects of home stress on six variables: moderate-intensity physical activity, the number and proportion of vegetable portions, the desire to eat, and proportions of orange and red chosen to color body silhouettes (Table 3). When comparing low and high home stress levels, large *ES* were observed for the number of vegetable portions and the proportion of orange-colored silhouettes. Moderate *ES* were found for moderate-intensity physical activity, proportion of red-colored silhouettes, and proportion of vegetables. Participants in the low-stress group engaged in less physical activity, consumed a greater number of vegetable portions, and had a higher proportion of orange-colored silhouettes and a lower proportion of red-colored silhouettes compared to the high-stress group. The *ES* between low and medium home stress for eating desire was large, with participants in the low-stress group expressing less desire to eat than the medium-stress group. Comparing medium and high home stress levels, large *ES* were observed for the proportion of orange-colored silhouettes, desire to eat, and number and proportion of vegetables. The *ES* was medium for the proportion of red-colored silhouettes. Medium-stress participants chose orange more often, had a greater desire to eat, consumed more vegetables, and had a higher proportion of orange-colored silhouettes compared to high-stress participants.

### 3.5. Effect of Work Stress Level on Individual, Behavioral, and Emotional Characteristics

The level of work stress had significant effects on six variables: age, number of alcohol and dairy product portions, proportion of orange-colored silhouettes, positive emotions, and proportion of vegetables in the diet (Table 4). Large *ES* were observed between low and high work stress levels for age, number of alcohol portions, proportion of orange-colored silhouettes, and proportion of vegetables. The *ES* was medium for the number of dairy product portions. Participants in the low-stress group were older, consumed fewer dairy product portions, had a higher proportion of vegetables, and chose the orange silhouette more often compared to the high-stress group. Comparing low and medium work stress levels, large *ES* were observed for positive emotions and proportion of vegetables, and a medium *ES* for orange-colored silhouettes. The low-stress group experienced more intense positive emotions, consumed a higher percentage of vegetables, and chose a higher proportion of orange-colored silhouettes compared to the medium-stress group. Comparing medium and high work stress levels, the *ES* was large for age and medium for the number of dairy product portions, orange-colored silhouettes, and positive emotions. Participants with medium stress were older, consumed fewer dairy product servings, had a higher proportion of orange-colored silhouettes, and experienced less intense positive emotions compared to the high-stress group.

## 4. Discussion

Home stress was found to be the primary contributor to general stress, followed by work stress. Stress levels can vary depending on the context, with higher levels of stress often experienced at work compared to home or vice versa. In this study, on average, stress levels were lower at home. However, there was a correlation between the two types of stress. Previous research has found a positive correlation between job stress and home stress [27]. A meta-analysis also showed that employees with greater job stress, lower job control, and limited support experienced more interference between work and family life [28]. Work-related stressors can spill over into family life, while home-related stressors can disrupt work.

### 4.1. Home Stress

There was a positive correlation between home stress and the proportion of red-colored silhouettes as well as the consumption of dairy products, while an inverse correlation was found with the proportion of orange-colored silhouettes and the consumption of vegetables. The relationship between color and emotion has been established. Red is associated with both positive and negative concepts, while other colors like blue or pink are less commonly associated with negative concepts [29]. Red is known to be associated with sexual attractiveness, passion, love, danger, and aggression. Additionally, red can also be associated with anger. Therefore, it is reasonable to expect that individuals who experienced high levels of stress may opt for a red-colored silhouette to represent themselves. This choice is consistent with the perception that red signifies heightened arousal and a perception of aggression when faced with challenges that surpass one’s coping ability. Orange represents an exciting emotion, albeit with less intensity compared to red, and yellow is milder and more pleasant than orange. Gray is often associated with feelings of sadness, shame, or fear.

Vegetable consumption, desire to eat, orange and red silhouettes, and physical activity were found to differentiate the three levels of home stress. The *ES* for vegetable consumption was large, with low or moderate stress levels being associated with higher vegetable consumption compared to high-stress levels. Increased consumption of vegetables, which are rich in antioxidants and folate, may contribute to reducing stress levels and protecting against mental illness [30]. On the contrary, chronic perceived stress can also impact eating behaviors, leading to increased consumption of sweet and salty snacks and caffeinated drinks, as well as reduced intake of vegetables and fruits [28]. The desire to eat was significantly greater among participants with medium home stress compared to the other two groups. This finding is consistent with previous research [31] that indicated that individuals with moderate stress tend to maintain a general desire for food even after experiencing stress, whereas the high-stress group had less desire for food and showed signs of anhedonia, i.e., difficulty perceiving one’s own emotions. The moderate stress group also had a greater desire to eat compared to the low-stress group. People with a high level of self-control tend to experience less stress, lower levels of desire, and less goal conflict, indicating an anticipatory coping strategy to avoid problematic situations [32]. There were large and moderate *ES* for orange and red silhouettes between low, moderate, and high stress levels. Hence, lower levels of stress were correlated with a higher proportion of orange silhouettes and a lower proportion of red silhouettes. As observed before, the color red is linked to high stress. The high-stress home group engaged in more moderate-intensity physical activity than the low-stress group. However, it has been demonstrated that perceived stress tends to decrease when individuals are engaged in vigorous or moderate-intensity physical activity, highlighting their potential positive impact on stress reduction [33]. Household activities, which fall into the moderate physical activity class, may contribute to home-related stress if they are perceived as laborious or mentally burdensome [34].

### 4.2. Work Stress

The evaluation of work stress revealed significant differences in age, alcohol consumption, positive emotions, and tendencies related to the consumption of dairy products, the proportion of consumed vegetables, and the proportion of orange-colored silhouettes. The participants with low and moderate stress levels were found to be older than the highly stressed group, which is consistent with previous studies indicating that younger employees tend to experience greater work stress [8,35]. In the predominantly female sample of this study, highly and moderately stressed participants consumed less alcohol compared to the low-stress group, which is consistent with the lower level of alcohol problems observed in stressed women due to higher levels of social support [36]. The *ES* for the dairy product proportion was moderate between low, moderate, and high home stress levels. The most stressed people at work consumed more dairy products than the other participants, probably for the calming effect of substances derived from milk. Dairy product intake influences stress and mental health [37]. Furthermore, individuals in the low and high work-related stress groups reported experiencing positive emotions more intensely. It has been established that higher levels of stress are positively associated with negative emotions and negatively associated with positive affect [38]. In our study, high levels of stress were also associated with positive emotions. Folkman and Moskowitz (2004) reported that negative and positive emotions can co-occur throughout the stress process [39]. Viney et al. (1989) found a co-occurrence of positive and negative emotions in a sample of chronically ill men [40]. Although the negative emotions were more frequent in the chronically ill groups when compared to a healthy control group, the positive emotion of enjoyment was also more frequent in the chronically ill groups.

In line with the findings regarding home stress, the low work stress group exhibited higher proportions of vegetables and orange-colored silhouettes. This indicates a similar association between stress level and vegetable consumption and the preference for the color orange in both home and work, highlighting their potential importance across different environments. Consumption of fatty–salty–sugary products in the present study, which are often palatable, was not higher among stressed employees than among those with low stress levels, either at home or at work. This result was not consistent with other studies that have found a positive association between stress and consumption of palatable foods. Instead of palatable foods, it was the proportion of dairy products in the diet that tended to be higher among employees who were stressed at work. Two hypotheses can be put forward to explain this result: employees are aware of the dangers of consuming fatty–salty–sugary products, and dairy products may have beneficial effects on mood.

Despite providing novel findings, this study has limitations. The volunteers come from a voluntary, non-randomized sample. The male/female and normal weight/overweight/obese ratios were not balanced. As height and weight were self-reported, they may not be entirely reliable. The cross-sectional nature of the data limits our ability to establish causal relationships, and larger and more balanced samples in terms of gender and weight status could provide clearer effects.

## 5. Conclusions

Both stress at home and at work contributed to stress in general, with a greater contribution from stress at home than at work. Different behavioral and perceptual variables were associated with one or another location where stress was evaluated. Certain foods and emotions were linked to stress recorded at home, while individual characteristics were associated with stress at work. These variables can result from stress, lead to a reduction in stress, or reinforce the perception of stress. The participants in our study who see themselves more frequently in red and those who consume more dairy products in their diet but fewer servings of vegetables than the other participants are more at risk of suffering from stress at home. Age seems to be protective, while high body fat is a marker of stress at work. Younger people who are overweight or obese may be more at risk of suffering work-related stress. However, our cross-sectional study is unable to establish causal conclusions regarding the impact of stress on behavior and perceptions due to the absence of temporal elements in the design, as well as the fact that the presumed cause was not assessed before the hypothesized effect.

To investigate deeper into the causal relationship, such as that between a diet abundant in naturally antioxidant-rich vegetables and the reduction in stress, it would be necessary to undertake an intervention study spanning several months. This study would involve the regular measurement of stress before, during, and after the intervention.

## Figures and Tables

**Figure 1 ijerph-22-01390-f001:**
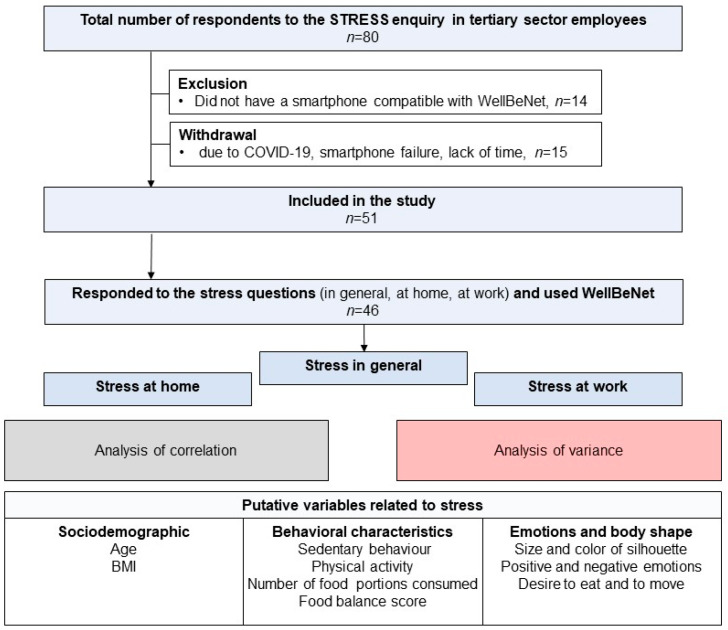
Flow chart of employees’ inclusion.

**Figure 2 ijerph-22-01390-f002:**
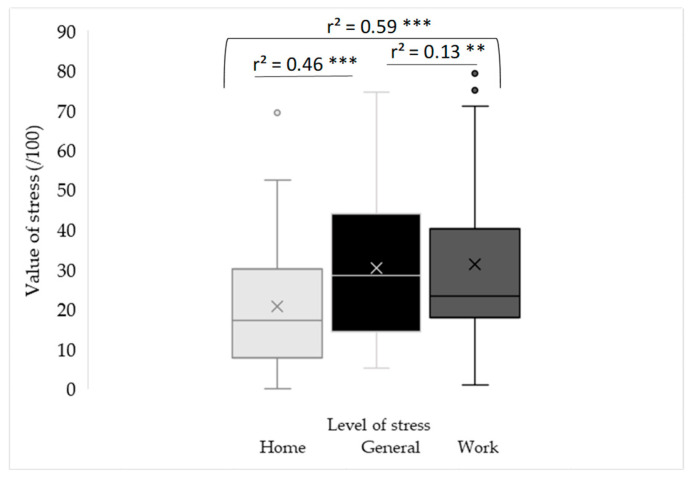
Distribution of stress values and contribution of stress at home and at work to stress in general (whisker chart). X: mean value, °: outlier, lower and upper lines outside the box: minimum and maximum data, excluding outliers, lower and upper lines in the box: 25th and 75th percentiles, line in the box: median (50th percentile). Level of significance: **: *p* < 0.01, ***: *p* < 0.001.

**Figure 3 ijerph-22-01390-f003:**
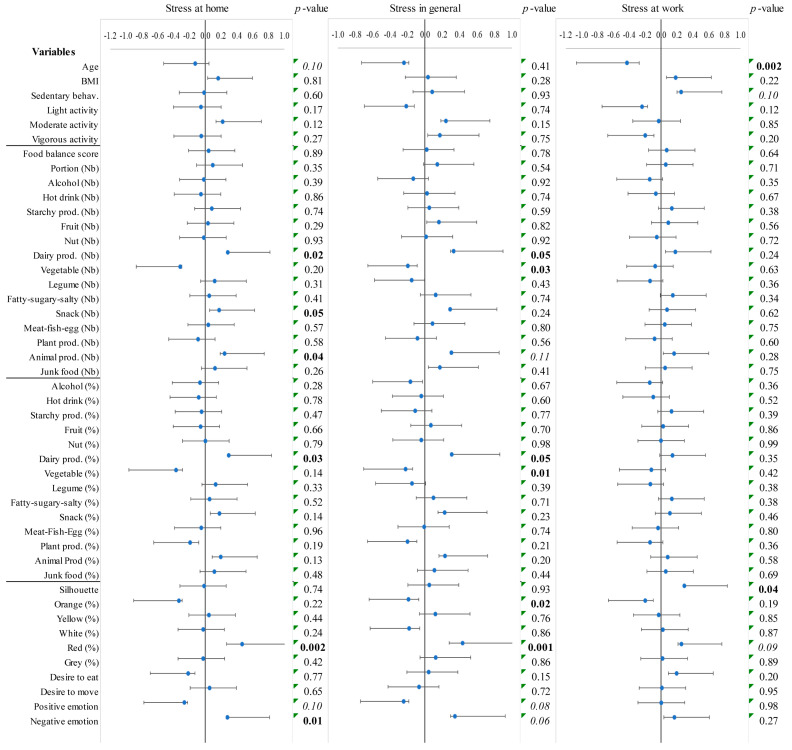
Linear relationships between stress and individual, behavioral, and emotional characteristics. Coefficients of correlation and their 95% confidence intervals. *p*-values in bold indicate a probability lower than 0.05, those in italics indicate a trend with a probability lower than or equal to 0.10.

**Table 1 ijerph-22-01390-t001:** Examples of food portion.

Food	Example of Portion
Alcohol	1 aperitif glass (strong alcohol) Cider, beer, and wine (a 25 cL glass)
Hot drinks	1 cup/bowl/glass of tea, herbal tea, coffee, infusion
Starchy products	1 slice of bread 1/3 plate of corn, rice, wheat pasta, semolina 3 to 5 medium-sized potatoes 1/4 bowl of breakfast cereals 2 rusks
Fruits	1 piece of fruit (100 to 150 g) 2 small fruits (50–75 g) 1 small bowl of fruit (fruit salad, fruit in syrup, compote, etc.) 1 handful of fruit (strawberries, cherries, raspberries, redcurrants, etc.)
Nuts	1 handful of walnuts, hazelnuts, almonds, pistachios
Dairy products	1 yogurt (125 g) 1 portion of cheese (30–50 g) 2 Petit Suisse cheeses (120 g) 1 glass/bowl/cup of milk (100 to 125 g) 1 small bowl of white cheese (100 g)
Vegetables	1/3 of a plate of cooked or raw vegetables (150 to 250 g)
Legumes	2 tablespoons of lentils, peas, beans, cooked flageolet beans (50 g)
Fatty-salty-sugary products	1 slice of cake 1 teaspoon of jam 1 serving of restaurant butter (10 g) 1 cone, small pot, scoop of ice cream 1 medium serving (fast food) of French fries 1 teaspoon of sauce (ketchup, mayonnaise, barbecue, pesto, vinaigrette, etc.)
Snacks	1 bowl of breakfast cereal (sweet, chocolate, etc.) 1 can of soda 1 small handful of candy 1 cookie 1 chocolate bar 1 pastry 1 small bag of chips (25 g) 1 handful of crackers 1/4 pizza, industrial quiche (28 cm diameter) 1 small pizza or individual industrial quiche 1 industrial crepe 1 hamburger (standard) 1 industrial ready meal Cold cuts: 2 thin slices of sausage, salami 3–4 slices of cervelat sausage, dry sausage 1 slice of ham (white, raw) 2 tablespoons of bacon bits 1/2 slice of pâté and rillettes
Meat-Fish-Eggs	1 cutlet 2 eggs 1 chicken thigh 1 fish steak 1 hamburger patty 1/3 of a seafood platter

**Table 2 ijerph-22-01390-t002:** Description of the volunteer sample.

Weight Status	Sex	Size	Age (year)	BMI (kg·m^−2^)	Executive	Non-Executive
NW	W	19	48.3 (5.6)	21.9 (2.1)	10	9
OW	W	11	46.5 (3.4)	26.6 (1.8)	3	8
OB	W	11	48.7 (4.9)	37.7 (10.3)	3	8
NW	M	5	52.0 (7.2)	23.7 (1.3)	3	2
OW	M	4	47.0 (3.6)	27.1 (1.4)	1	3
OB	M	1	46.0 (.)	31.0 (.)	0	1

NW: normal-weight, OW: overweight, OB: obese, W: women, M: men; . Since there is only one volunteer in the obese men group, we cannot calculate the standard deviation.

**Table 3 ijerph-22-01390-t003:** Mean comparison of individual, behavioral, and emotional characteristics between the three home stress levels in a sample of French tertiary sector employees.

Home Stress	Stress Level			*p*-Value ^f^	*ES* ^g^		
	Low	Medium	High		Low-Medium	Low-High	Medium-High
Size, no.	*n* = 15	*n* = 15	*n* = 16			**stress**	
Stress limit	≤10	]10–25]	>25				
Age, year	48.7 ± 3.8	47.5 ± 5.8	48.8 ± 5.6	0.76	0.09	−0.02	−0.23
BMI, kg/m^2^	27.3 ± 5.9	26.0 ± 5.3	29.2 ± 11.3	0.90	0.23	−0.21	−0.35
Home stress ^a^	4.9 ± 3.3	17.5 ± 4.3	40.0 ± 11.9	**<0.0001**	−3.29	−3.96	−2.48
**Physical activity ^b^**							
Sedentary, %	82.3 ± 8.0	80.5 ± 12.6	79.7 ± 11.4	0.81	0.17	0.26	0.07
Light, %	12.2 ± 6.8	14.4 ± 11.0	13.0 ± 10.4	0.94	−0.24	−0.09	0.13
Moderate, %	3.6 ± 2.4	4.1 ± 2.8	6.2 ± 4.1	*0.07*	−0.19	−0.77	−0.59
Vigorous, %	1.8 ± 2.5	1.0 ± 1.6	1.1 ± 1.6	0.44	0.38	0.34	−0.06
**Diet ^c^**							
Portion, no.	12.1 ± 3.3	13.7 ± 3.6	12.5 ± 4.1	0.49	−0.46	−0.11	0.31
Alcohol, no.	0.6 ± 0.6	0.4 ± 0.3	0.6 ± 1.0	0.46	0.42	0.00	−0.27
Hot drink, no.	1.6 ± 0.9	2.0 ± 0.9	1.6 ± 1.0	0.48	−0.44	0.00	0.42
Starchy prod, no.	2.7 ± 1.0	2.8 ± 0.9	2.6 ± 1.4	0.91	−0.11	0.08	0.17
Fruit, no.	1.4 ± 0.8	1.5 ± 0.6	1.0 ± 1.1	0.27	−0.14	0.41	0.56
Nut, no.	0.4 ± 0.5	0.5 ± 0.7	0.5 ± 0.7	0.61	−0.16	−0.16	0.00
Dairy prod, no.	1.4 ± 0.9	1.5 ± 0.8	1.8 ± 0.9	0.48	−0.12	−0.44	−0.35
Legume, no.	0.09 ± 0.1	0.3 ± 0.5	0.4 ± 1.2	0.40	−0.58	−0.36	−0.11
Vegetable, no.	1.7 ± 0.7	1.9 ± 0.8	1.1 ± 0.7	**0.02**	−0.27	0.86	1.07
Fatty–salty–sugary prod, no.	0.8 ± 0.8	0.7 ± 0.7	0.8 ± 0.8	0.97	0.13	0.00	−0.13
Snack, no.	0.4 ± 0.5	0.8 ± 0.8	0.8 ± 1.0	0.20	−0.60	−0.50	0.00
Meat–Fish–Egg, no.	1.2 ± 0.6	1.2 ± 0.7	1.1 ± 0.5	0.94	0.00	0.18	0.17
Plant prod, no.	3.6 ± 1.5	4.3 ± 1.8	3.2 ± 1.3	0.15	−0.42	0.29	0.70
Animal prod, no.	2.6 ± 1.0	2.8 ± 1.1	2.9 ± 1.2	0.73	−0.19	−0.21	−0.08
Junk food, no.	1.6 ± 1.2	1.9 ± 1.4	2.2 ± 1.5	0.48	−0.23	−0.44	−0.21
Food balance score	5.6 ± 1.2	6.0 ± 1.2	5.4 ± 1.1	0.45	−0.33	0.17	0.52
**Proportion of one food category in the diet ^d^**							
Alcohol, %	4.3 ± 6.1	2.5 ± 2.3	4.9 ± 7.6	0.82	0.39	−0.09	−0.42
Hot drink, %	13.8 ± 7.9	14.4 ± 4.5	12.7 ± 8.4	0.80	−0.09	0.13	0.25
Starchy prod, %	22.1 ± 5.6	21.0 ± 7.1	20.0 ± 7.1	0.66	0.17	0.33	0.14
Fruit, %	11.6 ± 6.8	10.9 ± 3.8	8.0 ± 8.0	0.29	0.13	0.48	0.46
Nut, %	3.1 ± 3.9	3.5 ± 3.8	4.3 ± 4.9	0.73	−0.10	−0.27	−0.18
Dairy prod, %	10.9 ± 5.9	11.0 ± 5.4	14.9 ± 7.8	0.17	−0.02	−0.58	−0.58
Legume, %	0.8 ± 0.9	2.1 ± 3.5	3.6 ± 9.4	0.35	−0.51	−0.41	−0.21
Vegetable, %	14.1 ± 11.1	14.5 ± 6.3	9.5 ± 5.6	**0.05**	−0.04	0.53	0.84
Fatty–salty–sugary prod, %	5.8 ± 5.4	4.6 ± 4.1	5.8 ± 4.9	0.76	0.25	0.00	−0.26
Snack, %	2.7 ± 3.3	5.9 ± 5.2	6.6 ± 9.3	0.24	−0.73	−0.55	−0.09
Meat–Fish–Egg, %	10.7 ± 6.7	9.5 ± 5.1	9.7 ± 4.6	0.81	0.20	0.18	−0.04
Plant prod, %	29.5 ± 11.1	31.0 ± 10.3	25.5 ± 8.8	0.30	−0.15	0.40	0.59
Animal prod, %	21.6 ± 7.2	20.5 ± 7.9	24.6 ± 10.4	0.42	0.15	−0.33	−0.44
Junk food, %	12.9 ± 9.3	13.1 ± 7.2	17.2 ± 10.8	0.34	−0.02	−0.43	−0.44
**Perception ^e^**							
Body shape	4.9 ± 1.2	4.4 ± 0.9	4.6 ± 1.1	0.49	0.47	0.26	−0.20
Yellow, %	12.0 ± 18.7	13.3 ± 14.9	19.4 ± 30.6	0.86	−0.08	−0.29	−0.25
White, %	46.9 ± 43.4	38.4 ± 41.0	48.7 ± 45.3	0.66	0.20	−0.04	−0.24
Orange, %	31.7 ± 33.9	38.5 ± 38.7	5.4 ± 10.5	*0.07*	−0.19	1.06	1.19
Red, %	6.1 ± 10.3	7.5 ± 13.9	23.5 ± 35.6	*0.08*	−0.11	−0.65	−0.58
Grey, %	2.6 ± 5.7	2.1 ± 7.7	2.8 ± 6.0	0.14	0.07	−0.03	−0.10
Desire to eat	32.9 ± 14.9	47.4 ± 19.4	31.7 ± 15.1	**0.02**	−0.84	0.08	0.91
Desire to move	39.2 ± 15.4	40.4 ± 17.8	42.5 ± 21.1	0.87	−0.02	−0.12	−0.09
Positive emotion	2.9 ± 1.3	3.2 ± 0.5	2.4 ± 1.2	0.19	−0.61	0.16	0.86
Negative emotion	1.2 ± 0.7	2.0 ± 1.2	1.2 ± 0.9	0.12	−0.81	0.00	0.76

^a^ Perceived stress at work in the evening. Score may range from 0 to 100. Values are mean ± SD. ^b^ Physical activity was assessed using smartphone accelerometers throughout the participants’ waking period. By analyzing the intensity of the accelerometry signals, each minute was categorized into one of four levels: immobility, light intensity, moderate intensity, and vigorous intensity. The duration of time spent in each category was then expressed as a percentage of the total recording time. ^c^ Dietary intake was evaluated by quantifying the number of food portions consumed across 12 predefined categories. Values are mean ± SD. ^d^ The proportion of each food category was calculated by dividing the number of portions for each specific food category by the total number of portions consumed across all 12 categories. ^e^ Each evening, the participants associated their body shape with one of the nine silhouettes defined by Stunkard et al. (1983) and chose one color for their silhouette [20]. The proportion of a specific color was calculated by dividing the number of occurrences of that color by the total number of colors. Each evening, the participants rated their desire to eat and to move on 100-point scales, and one or several positive and negative emotions defined by Scherer (2005) [24] on five-point scales. Means and SD were calculated based on the total number of evaluated emotions. ^f^ Determined by GLM; *p* < 0.10 considered as a trend (italics), *p* < 0.05 considered as significant (bold). ^g^ *ES*: effect size.

**Table 4 ijerph-22-01390-t004:** Mean comparison of individual, behavioral, and emotional characteristics between the three work stress levels in a sample of French tertiary sector employees.

Work Stress		Stress Level		*p*-Value ^f^	*ES* ^g^		
	Low	Medium	High		Low-Medium	Low-High	Medium-High
Size nb	*n* = 15	*n* = 16	*n* = 15			**stress**	
Stress limit	≤19	]19–32]	>32				
Age y	49.9 ± 6.2	49.4 ± 4.7	45.7 ± 3.9	**0.04**	0.09	0.81	0.85
BMI kg/m^2^	26.4 ± 3.5	27.0 ± 5.7	29.2 ± 11.3	*0.08*	−0.13	−0.59	−0.25
Work stress ^a^	13.0 ± 5.8	24.8 ± 4.4	56.1 ± 16.0	**<0.0001**	−2.56	−3.69	−2.71
**Physical activity ^b^**							
Sedentary %	78.2 ± 8.8	80.3 ± 12.5	83.9 ± 8.5	0.38	−0.19	−0.66	−0.33
Light %	15.5 ± 7.1	13.3 ± 10.8	10.9 ± 8.5	0.38	0.24	0.59	0.25
Moderate %	4.4 ± 2.3	5.3 ± 4.3	4.2 ± 2.6	0.66	−0.27	0.08	0.31
Vigorous %	1.8 ± 1.2	1.0 ± 1.6	1.0 ± 1.1	0.62	0.56	0.69	0.00
**Diet ^c^**							
Portion nb	11.8 ± 3.3	13.2 ± 3.9	13.3 ± 4.2	0.45	−0.39	−0.40	−0.02
Alcohol nb	0.9 ± 0.6	0.4 ± 0.3	0.3 ± 0.3	**0.02**	1.07	1.26	0.33
Hot drink nb	1.5 ± 1.0	2.1 ± 1.2	1.6 ± 0.9	0.19	−0.54	−0.11	0.47
Starchy prod nb	2.4 ± 0.8	2.7 ± 1.3	3.0 ± 1.1	0.39	−0.28	−0.62	−0.25
Fruit nb	1.1 ± 0.6	1.4 ± 0.8	1.4 ± 1.0	0.47	−0.56	−0.36	0.00
Nut nb	0.4 ± 0.5	0.6 ± 0.8	0.6 ± 0.6	0.26	−0.30	−0.36	0.00
Dairy prod nb	1.3 ± 0.9	1.5 ± 0.9	2.0 ± 0.9	*0.09*	−0.22	−0.78	−0.56
Legume nb	0.4 ± 1.3	0.4 ± 1.3	0.1 ± 0.1	0.74	0.00	0.33	0.32
Vegetable nb	1.8 ± 0.7	1.4 ± 0.9	1.5 ± 0.6	0.29	0.49	0.46	−0.13
Fatty–salty–sugary prod nb	0.7 ± 0.7	0.9 ± 0.7	0.9 ± 0.8	0.38	−0.14	−0.27	0.00
Snack nb	0.4 ± 0.5	0.8 ± 1.0	0.7 ± 0.9	0.18	−0.50	−0.41	0.10
Meat–Fish–Egg nb	1.2 ± 0.5	1.1 ± 0.6	1.2 ± 0.7	0.90	0.18	0.00	−0.15
Plant prod nb	3.5 ± 1.3	3.7 ± 1.1	3.7 ± 1.7	0.94	−0.17	−0.13	0.00
Animal prod nb	2.5 ± 1.0	2.7 ± 1.2	3.2 ± 1.3	0.21	−0.18	−0.60	−0.40
Junk food nb	1.8 ± 1.0	2.0 ± 1.5	1.9 ± 1.3	0.93	−0.27	−0.09	0.07
Food balance score	5.5 ± 1.2	5.6 ± 1.1	6.0 ± 1.3	0.45	−0.09	−0.40	−0.33
**Proportion of one food category in the diet ^d^**							
Alcohol %	7.1 ± 6.1	2.3 ± 5.9	2.5 ± 2.5	0.31	0.80	1.02	−0.04
Hot drink %	13.2 ± 7.4	15.8 ± 7.4	11.7 ± 6.9	0.28	−0.35	0.21	0.57
Starchy prod %	20.7 ± 4.2	20.0 ± 6.9	22.3 ± 6.1	0.65	0.12	−0.31	−0.35
Fruit %	9.0 ± 6.7	11.4 ± 7.3	10.0 ± 7.6	0.61	−0.34	−0.14	0.19
Nut %	3.4 ± 4.5	2.9 ± 4.8	4.5 ± 4.4	0.21	0.11	−0.25	−0.35
Dairy prod %	10.8 ± 6.4	11.5 ± 5.0	14.9 ± 7.6	0.20	−0.12	−0.58	−0.53
Legume %	1.7 ± 3.4	3.9 ± 9.7	0.9 ± 0.8	0.78	−0.30	0.32	0.43
Vegetable %	15.4 ± 4.6	10.9 ± 6.4	11.5 ± 4.3	0.10	0.80	0.88	−0.11
Fatty–salty–sugary prod %	4.3 ± 5.0	5.7 ± 4.2	6.2 ± 4.9	0.39	−0.30	−0.38	−0.11
Snack %	2.9 ± 3.5	6.6 ± 8.8	5.8 ± 6.6	0.36	−0.55	−0.55	0.10
Meat–Fish–Egg %	11.4 ± 6.1	8.6 ± 4.4	9.9 ± 5.4	0.37	0.53	0.26	−0.26
Plant prod %	29.5 ± 7.8	29.2 ± 7.5	26.9 ± 9.9	0.76	0.04	0.29	0.26
Animal prod %	22.2 ± 6.5	20.1 ± 6.9	24.7 ± 10.5	0.35	0.31	−0.29	−0.52
Junk food %	14.4 ± 6.8	14.7 ± 9.1	14.4 ± 8.1	0.99	−0.04	0.00	0.03
**Perception ^e^**							
Body shape	4.2 ± 1.0	4.5 ± 1.1	5.0 ± 1.2	0.18	−0.28	−0.72	−0.44
Yellow %	13.7 ± 13.3	18.4 ± 28.7	12.9 ± 20.1	0.96	−0.23	0.02	0.22
White %	29.2 ± 46.9	57.3 ± 43.3	48.3 ± 43.6	0.27	−0.62	−0.42	0.21
Orange %	46.6 ± 38.1	11.5 ± 22.3	14.8 ± 21.9	*0.07*	1.13	1.02	−0.15
Red %	7.3 ± 9.9	9.3 ± 25.7	21.7 ± 29.6	0.21	−0.10	−0.65	−0.45
Gray %	2.1 ± 5.6	3.4 ± 8.1	2.0 ± 5.4	0.99	−0.19	0.02	0.20
Desire to eat	35.6 ± 14.8	32.7 ± 18.9	42.6 ± 20.7	0.29	0.17	−0.39	−0.50
Desire to move	43.1 ± 14.3	36.3 ± 14.8	43.9 ± 19.5	0.39	0.47	−0.05	−0.44
Positive emotion	3.4 ± 1.3	2.2 ± 1.2	2.9 ± 0.8	**0.05**	0.88	0.37	−0.68
Negative emotion	1.6 ± 0.7	1.2 ± 1.0	1.4 ± 0.9	0.60	0.46	0.25	−0.21

^a^ Perceived stress at work in the evening. Score may range from 0 to 100. Values are mean ± SD. ^b^ Physical activity was assessed using smartphone accelerometers throughout the participants’ waking period. By analyzing the intensity of the accelerometry signals, each minute was categorized into one of four levels: immobility, light intensity, moderate intensity, and vigorous intensity. The duration of time spent in each category was then expressed as a percentage of the total recording time. ^c^ Dietary intake was evaluated by quantifying the number of food portions consumed across 12 predefined categories. Values are mean ± SD. ^d^ The proportion of each food category was calculated by dividing the number of portions for each specific food category by the total number of portions consumed across all 12 categories. ^e^ Each evening, the participants associated their body shape with one of the nine silhouettes defined by Stunkard et al. (1983) and chose one color for their silhouette [20]. The proportion of a specific color was calculated by dividing the number of occurrences of that color by the total number of colors. Each evening, the participants rated their desire to eat and to move on 100-point scales, and one or several positive and negative emotions defined by Scherer (2005) [24] on five-point scales. Means and SD were calculated based on the total number of evaluated emotions. ^f^ Determined by GLM; *p* < 0.10 considered as a trend (italics), *p* < 0.05 considered as significant (bold). ^g^ *ES*: effect size.

## Data Availability

The data presented in this study are available from the corresponding author upon reasonable request.

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
