# Peer review of "Associations Between Stress Level, Environment, and Emotional and Behavioral Characteristics in Service Sector Employees"

_ijerph, 2025, doi:10.3390/ijerph22091390_

Round 1

Reviewer 1 Report

Comments and Suggestions for Authors

Whilst this is a solidly written paper, with appropriate measures used appropriately, the authors have not made a strong enough case for how this paper adds to the extant knowledge. This is an interesting paper, but it would benefit from a more critical approach and tone in both the introductory section and the discussion to further elaborate how this paper adds (or indeed negates) previous research in this area. Although I note that a bespoke device is used to gather this research, I am not sure what the influence of that app is, if any, from reading this paper. 

So in summary, this paper is technically strong and well written, but it needs significant work in order to argue its contribution to the field. 

Author Response

Reviewer 1:

Comments 1: Whilst this is a solidly written paper, with appropriate measures used appropriately, the authors have not made a strong enough case for how this paper adds to the extant knowledge. This is an interesting paper, but it would benefit from a more critical approach and tone in both the introductory section and the discussion to further elaborate how this paper adds (or indeed negates) previous research in this area. Although I note that a bespoke device is used to gather this research, I am not sure what the influence of that app is, if any, from reading this paper. So in summary, this paper is technically strong and well written, but it needs significant work in order to argue its contribution to the field. 

Response 1: Thank you for your comments and compliments. Some knowledge precedes the results of our study: stress is associated with a decrease in leisure-time physical activity and an increase in screen time. It increases the consumption of palatable foods and decreases the consumption of fruits and vegetables. Chronic stress can increase weight gain. Stress is greater in women than in men and greater in young people than in people over 60. Stress can exist at work and at home due to the nature of human relationships at work or within the family circle, hierarchical relationships, responsibilities, recognition at work, sedentary behavior at work...

Our study took a comprehensive approach to stress by collecting multiple data points on physical activity, diet, and emotions in different contexts (work, home, in general) and thus supplemented existing knowledge. The originality of our work lies in the study of the specific population of sedentary employees. In this population, the closer link between general stress and stress at home than between general stress and stress at work was also original. Even if stress at work is higher than stress at home, it is less associated with general stress. The explanatory variables for stress at home were not the same as those for stress at work. At home, stress was correlated with the perception of one’s silhouette in red, a higher consumption of dairy products and a lower consumption of vegetables. At work, stress was mainly associated with age and body shape. This different type of variables observed depending on the context was original.

The objective assessment of sedentary behavior and physical activity using algorithms previously developed specifically for normal weight and overweight adults gave original evaluation of behaviors (algorithms validated and published in several articles since 2013). We have statistical data on physical activity and sedentary lifestyles. We observe that employees who were stressed at home engaged in moderate physical activity half as much as those who were least stressed. This difference in behavior is not observed between the most stressed employees and those who are least stressed at work. We have quantified data on diet and perceptions. Vegetable consumption was higher among employees who experienced little or moderate stress at home compared to employees who experienced high stress. This difference in consumption between the three levels of stress at work was less pronounced. Only a trend can be observed in the distribution of vegetables among other food categories. These results corroborate those of other studies. However, consumption of fatty-salty-sugary products, which are often palatable, was not higher among stressed employees than among those with low stress levels, either at home or at work. This result was not consistent with other studies that have found a positive association between stress and consumption of palatable foods. Instead of palatable foods, it was the proportion of dairy products in the diet that tended to be higher among employees stressed at work. Two hypotheses can be put forward to explain this result: employees are aware of the dangers of consuming fatty-salty-sugary products, and dairy products may have beneficial effects on mood. Regarding perceptions, it is uncommon to ask participants to associate a color with their silhouette and for researchers to make a connection between the color of the silhouette and the level of stress. The colors orange and red made sense, as red was more frequently chosen when employees were stressed and orange was less frequently chosen. The color red is associated with strong excitement, such as anger or passion. Orange is a softer color that indicates a moderate emotion. Participants sometimes had difficulty verbalizing and assessing the intensity of their emotions. Choosing a color seemed simpler and less thoughtful.

Entering the number of food portions immediately after eating can be done very quickly using the app, which is an advantage in limiting forgetfulness or rushing through the task.

Some of the above comments have been incorporated into the discussion.

Reviewer 2 Report

Comments and Suggestions for Authors

The paper appears to be a report documenting the feasibility of using the "WellBeNet application" to assess stress and its relationship with several behavioral factors (diet, physical activity) and emotions. The Authors did not provide a purpose for the study. Instead, they wrote: "We hypothesized that the level of stress could vary depending on the context and be associated with various behaviors. This is not a hypothesis, as these relationships have been repeatedly demonstrated."

The study group is small, and there is no study protocol described. The text further indicates that participants recorded their feelings and behaviors for a week. I disagree with the statement: "This study aimed to objectively assess physical activity, self-reported stress levels, emotions, and food choices in a random sample of tertiary employees aged 40-60 years" (Line 81-82). This type of study allows only subjective assessment. Furthermore, a random sample doesn't seem to apply to this type of study, and if so, how was the randomization conducted? In my opinion, the study simply included volunteers who participated in the study.

To properly evaluate this study, a detailed description of the "WellBeNet" application and its components (eMouve, NutriQuantic, and EmoSens) is necessary. Most readers do not have access to this application, so it is difficult to assess its value, and not all the references indicated are publicly available, e.g.,number 17.

The description of the study group is very limited; only the mean age, without even a standard deviation, and the number of overweight and obese individuals are provided, but without a BMI. It's not even clear what these ranges were or what criteria were used for classification. There is a lack of data on education and family situation, which significantly affects the level of stress and what the Authors mentioned  in the Introduction.

There's also no description of the work they performed, even if it was physical or mental work. A description of the entire study group, including the division into men and women, should be provided.

The stress assessment is also very rough; it's unclear what factors at work and at home were the source of stress.

When describing diets, it should be clarified what the authors understand by “portion”

As can be seen from Tables 1 and 2, the Authors did not take gender into account in their analyses, which is an incorrect approach, as the literature shows that both the sense of stress at work and at home differs depending on gender [Mensah A. Job Stress and Mental Well-Being among Working Men and Women in Europe: The Mediating Role of Social Support. Int J Environ Res Public Health. 2021 Mar 3;18(5):2494. doi: 10.3390/ijerph18052494]

It would be worthwhile to describe in the Methods a method for assessing body shape using one of the nine silhouettes defined by Stunkard et al. (1983), as this literature is not widely available. They selected one of nine 127 silhouettes defined by Stunkard et al. (1983). It is also not clear if it was nine or 5  silhouettes: orange, yellow, red, gray, or white (Line 128, ). The Authors do not explain the meaning of the specific colors.

Moreover, the next cited publication by Nummenmaa et al. (2014) does not indicate that this method can be used to assess emotions after a full day. These emotions are associated rather with specific words, stories, movies, and faces.

In "Limitations" more shortcomings should be taken into account

Author Response

Reviewer 2:

Comment 1: The paper appears to be a report documenting the feasibility of using the "WellBeNet application" to assess stress and its relationship with several behavioral factors (diet, physical activity) and emotions. The Authors did not provide a purpose for the study. Instead, they wrote: "We hypothesized that the level of stress could vary depending on the context and be associated with various behaviors. This is not a hypothesis, as these relationships have been repeatedly demonstrated."

Response 1: Thank you for your comments and request for clarification. We did indeed use WellBeNet to study the relationships between stress, behaviors, and perceptions. The purpose of the study is to first examine the relationships between general stress and stress at work and at home, then between stress and behavioral and perceptual variables depending on the context. The purpose of the study is now specified in the introduction. The sentence “We hypothesized that the level of stress could vary depending on the context and be associated with various behaviors” has been deleted.

Comment 2: The study group is small, and there is no study protocol described. The text further indicates that participants recorded their feelings and behaviors for a week. I disagree with the statement: "This study aimed to objectively assess physical activity, self-reported stress levels, emotions, and food choices in a random sample of tertiary employees aged 40-60 years" (Line 81-82). This type of study allows only subjective assessment. Furthermore, a random sample doesn't seem to apply to this type of study, and if so, how was the randomization conducted? In my opinion, the study simply included volunteers who participated in the study.

Response 2: The group size is indeed modest, but statistical effects are observable. Physical activity and sedentary behaviors were measured objectively using accelerometers built into smartphones and validated algorithms that process accelerometry data (published between 2013 and 2017). Self-reported stress levels, emotions, and food choices were subjective assessments. The participants of the study come from voluntary sample of tertiary employees aged 40-60 years and not from a random sample. These clarifications and corrections have been made in the revised version.

Comment 3: To properly evaluate this study, a detailed description of the "WellBeNet" application and its components (eMouve, NutriQuantic, and EmoSens) is necessary. Most readers do not have access to this application, so it is difficult to assess its value, and not all the references indicated are publicly available, e.g.,number 17.

Response 3: The application is only accessible to researchers and volunteers participating in research projects led by INRAE. However, several publications refer to how it works. For physical activity and sedentary lifestyles, please see the publications below, particularly Guidoux et al. 2014, which explains how the physical activity/ sedentary behavior assessment algorithm was designed.

                        Rousset S, Coyault Abele D, Benoit M, Zemni R, Lacomme P, Fleury G. (2020). Spontaneous Physical Activity and Sedentary Patterns Analyzed in a General Population of Adults by the eMouve Application. Springer Nature Switzerland AG 2020 T. Ahram et al. (Eds.) 1152, 363–368, doi: https://doi.org/10.1007/978-3-030-44267-5_54.

                        Paris L, Guidoux R, Saboul D, Duclos M, Boirie Y, Rousset S. (2019). Comparison of Active and Sedentary Bout Lengths in Normal and Overweight Adults using eMouveRecherche. International Journal of Sports and Exercice Medicine 5, (11), 151-160, doi: 10.23937/2469-5718/1510151.

                        Rousset S, Guidoux R, Paris L, Farigon N, Boirie Y, Lacomme P, Phan R, Ren L, Saboul D & Duclos M. (2018). eMouveRecherche: the first scientific application to promote light-intensity activity for the prevention of chronic diseases. Biology, Engineering and Medicine 3, (1), 1-6, doi: 10.15761/BEM.1000133.

                        Guidoux R, Duclos M, Fleury G, Lacomme P, Lamaudière N, Saboul D, Ren L & Rousset S. (2017). The eMouveRecherche application competes with research devices to evaluate energy expenditure, physical activity and still time in free-living conditions. Journal of Biomedical Informatics 69, 128-134, doi: 10.1016/j.jbi.2017.04.005.

                        Rousset S, Guidoux R, Paris L, Farigon N, Miolanne M, Lahaye C, Duclos M, Boirie Y & Saboul D. (2017). A novel smartphone accelerometer application for low-intensity activity and energy expenditure estimations in overweight and obese adults. Journal of Medical Systems 41, (117), 1-10, doi: 10.1007/s10916-017-0763-y.

                        Paris L, Duclos M, Guidoux R & Rousset S . (2016). Evaluation of physical activity and energy expenditure in overweight and obese adults. International Journal of Sports and Exercice Medicine 2, (3), 1-6.

                        Duclos M, Fleury G, Lacomme P, Phan R, Ren L, Rousset S . (2016). An acceleration vector variance based method for energy expenditure estimation in real-life environment with a Smartphone/Smartwatch integration. Expert Systems with Applications 63, 435-449, doi: 10.1016/j.eswa.2016.07.021.

                        Rousset S, Fardet A, Lacomme P, Normand S, Montaurier C, Boirie Y & Morio, B. (2015). Comparison of total energy expenditure assessed by two devices in controlled and free-living conditions. European Journal of Sport Science 15, (5), 391-399, doi: 10.1080/17461391.2014.949309.

                        Duclos M, Fleury G, Guidoux R, Lacomme P, Lamaudière N, Manenq P-H, Paris L, Ren L, Rousset S. (2015). Use of smartphone accelerometers and signal energy for estimating energy expenditure in daily-living conditions. Current Biotechnology 4, (1), 4-15, doi: http://www.eurekaselect.com/128913#.

                        Guidoux R, Duclos M, Fleury G, Lacomme P, Lamaudière N, Manenq P-H, Paris L, Ren L, Rousset S. (2014). A smartphone-driven methodology for estimating physical activities and energy expenditure in free living conditions. Journal of Biomedical Informatics 52, 271-278, doi: 10.1016/j.jbi.2014.07.009.

                        Guidoux R, Boualit R, Duclos M, Fleury G, Lacomme P, Lamaudière N, Rousset S. (2013). Conception d'une nouvelle fonction d'estimation de la dépense énergétique adaptée aux smartphones et aux conditions habituelles de vie. Nutrition clinique et métabolisme 27, S57.

For NutriQuantic, the number of food servings consumed in 11 food categories (alcohol, hot drink, starchy product, fruit, nut, dairy product, legume, vegetable, fatty-salty-sugary product, snack, and meat-fish-egg) could be reported by volunteers. Plant products include starchy foods, fruits, legumes, nuts and vegetables. Animal products include meat-fish-egg, dairy product and junk foods include alcohol, fatty-salty-sugary products and snack. A nutritional score was assigned to each food category (except hot drink): A nutritional score of 0 corresponds to a number of servings that is very far from the nutritional recommendations, 0.5 to a number that is slightly far from the recommendations, and 1 to a number of servings in line with the recommendations. The balance score is the sum of the individual scores. These details have been added to the revised version.

For EmoSens, the nine silhouettes (body shape) come from the work of Stunkard et al. (1983), the colors from the work of Nummenmaa et al. (2014), the emotions from the work of Scherer (2005) as mentioned in the publication. To quantify the desire to eat and to move, volunteers had to score their desire to eat and to move on a non-structured scales by sliding their finger along the scale. For a presentation of the application, please visit the website (https://activcollector.clermont.inra.fr/home) and watch the video. These details have been added to the revised version.

Comment 4: The description of the study group is very limited; only the mean age, without even a standard deviation, and the number of overweight and obese individuals are provided, but without a BMI. It's not even clear what these ranges were or what criteria were used for classification. There is a lack of data on education and family situation, which significantly affects the level of stress and what the Authors mentioned in the Introduction. There's also no description of the work they performed, even if it was physical or mental work. A description of the entire study group, including the division into men and women, should be provided.

Response 4: We have added the table below with the volunteers’ characteristics: mean and (standard deviation) for both age and BMI, and number of men and women, and number of employees according to professional status (executive or non-executive employee). All have intellectual office jobs without significant physical activity. We have no data on family situation.

Weight status

Sex

Size

Age (y)

BMI (kg.m-2)

Executive

Non-executive

NW

W

19

48.3 (5.6)

21.9 (2.1)

10

9

OW

W

11

46.5 (3.4)

26.6 (1.8)

3

8

OB

W

11

48.7 (4.9)

37.7 (10.3)

3

8

NW

M

5

52.0 (7.2)

23.7 (1.3)

3

2

OW

M

4

47.0 (3.6)

27.1 (1.4)

1

3

OB

M

1

46.0 (.)

31.0 (.)

0

1

NW: normal-weight, OW: overweight, OB: obese, W: women, M: men

A one-way analysis of variance did not show difference in age depending on weight status (F=1.02, p=0.37). Chi-square tests show that the number of executive employees was slightly higher among normal-weight individuals than among overweight individuals ( χ²= 2.83, p=0.09) or obese volunteers (χ²= 2.75, p=0.09). There was no difference in the number of executive and non-executive employees among overweight and obese volunteers.

Comment 5: The stress assessment is also very rough; it's unclear what factors at work and at home were the source of stress.

Response 5. The aim was to assess overall stress levels at the end of the day at work, at home, and in general. There was no in-depth investigation into the origin of stress. This experimental protocol was chosen to answer our research question, which was to establish links between these three types of stress and links between stress in two contexts with behavioral and perception variables in service sector employees. Furthermore, the longer the questionnaires, the less willing volunteers are to complete them due to lack of time and a feeling of intrusion. This can result in missing or incomplete data.  Moreover, in the publication that you have mentioned below (Mensah, 2021, Comment 7), the author used a single-item measure of job stress “do you experience stress in your work” and the response was rated on a Lickert scale (as we). The author mentioned that “the scale was validated with different health outcomes (Topp et al., 2015)”.

Comment 6: When describing diets, it should be clarified what the authors understand by “portion”

Response 6: For each food category, we provided examples of portion size. Please see the table below. This table has been added in the revised article.

Alcohol

1 aperitif glass (strong alcohol)

Cider, beer and wine (1 25cL glass)

Hot drinks

1 cup/bowl/glass of tea, herbal tea, coffee, infusion

Starchy products

1 slice of bread

1/3 plate of corn, rice, wheat pasta, semolina

3 to 5 medium-sized potatoes

¼ bowl of breakfast cereals

2 rusks

Fruits

1 piece of fruit (100 à 150g) 

2 small fruits (50-75g)

1 small bowl of fruit (fruit salad, fruit in syrup, compote, etc.)

1 handful of fruit (strawberries, cherries, raspberries, redcurrants, etc.)

Nuts

1 handful of walnuts, hazelnuts, almonds, pistachios

Dairy products

1 yogurt (125g)

1 portion of cheese (30-50g)

2 Petit Suisse cheeses (120g)

1 glass/bowl/cup of milk (100 to 125g)

1 small bowl of white cheese (100g)

Vegetables

1/3 of a plate of cooked or raw vegetables (150 to 250g)

Legumes

2 tablespoons of lentils, peas, beans, cooked flageolet beans (50g)

Fatty-salty-sugary products

1 slice of cake   

1 teaspoon of jam

1 serving of restaurant butter (10g)

1 cone, small pot, scoop of ice cream

1 medium serving (fast food) of French fries

1 teaspoon of sauce (ketchup, mayonnaise, barbecue, pesto, vinaigrette, etc.)

Snacks

1 bowl of breakfast cereal (sweet, chocolate, etc.)

1 can of soda

1 small handful of candy

1 cookie

1 chocolate bar

1 pastry   

1 small bag of chips (25g)

1 handful of crackers

¼ pizza, industrial quiche (28cm diameter)

1 small pizza or individual industrial quiche

1 industrial crepe

1 hamburger (standard)

1 industrial ready meal

Cold cuts:   

- 2 thin slices of sausage, salami

- 3-4 slices of cervelat sausage, dry sausage

- 1 slice of ham (white, raw)

- 2 tablespoons of bacon bits

- ½ slice of pâté and rillettes

Meat-Fish-Eggs

1 cutlet

2 eggs

1 chicken thigh   

1 fish steak

1 hamburger patty   

1/3 of a seafood platter

Comment 7: As can be seen from Tables 1 and 2, the Authors did not take gender into account in their analyses, which is an incorrect approach, as the literature shows that both the sense of stress at work and at home differs depending on gender [Mensah A. Job Stress and Mental Well-Being among Working Men and Women in Europe: The Mediating Role of Social Support. Int J Environ Res Public Health. 2021 Mar 3;18(5):2494. doi: 10.3390/ijerph18052494].

Response 7: This article shows that job stress negatively affects mental well-being among European workers (both men and women). The level of job stress is slightly greater among women than among men (men: 2.89 ± 1.16, women: 2.96 ± 1.12). This very minimal difference is observable in a population of 14,603 men and 15,486 women. Very large samples tend to transform small differences into statistically significant differences, even when they are clinically insignificant.  In our study, given the small sample size and the imbalance between the number of women and men, we chose not to take gender into account.

Comment 8: It would be worthwhile to describe in the Methods a method for assessing body shape using one of the nine silhouettes defined by Stunkard et al. (1983), as this literature is not widely available. They selected one of line 127 silhouettes defined by Stunkard et al. (1983). It is also not clear if it was nine or 5 silhouettes: orange, yellow, red, gray, or white (Line 128, ). The Authors do not explain the meaning of the specific colors. Moreover, the next cited publication by Nummenmaa et al. (2014) does not indicate that this method can be used to assess emotions after a full day. These emotions are associated rather with specific words, stories, movies, and faces.

Response 8: The silhouettes defined by Stunkard et al. (1983) are often used in research studies to allow volunteers to assess their body size. There are 9 of them. A recent publication presents these silhouettes:

Parzer V, Sjöholm K, Brix JM, Svensson P-A, Ludvik B, Taube M. Development of a BMI-Assigned Stunkard Scale for the Evaluation of Body Image Perception Based on Data of the SOS Reference Study. Obes Facts 2021;14:397–404 DOI: 10.1159/000516991

This reference has been added in the revised manuscript.

In our study, volunteers chose one silhouette from the nine proposed, then chose one color from the five proposed (orange, yellow, white, red, and gray). We asked them to choose a color that they associated with the silhouette corresponding to their body size. The choice of their color was made spontaneously without any information about the meaning of the color. The color red is often associated with a high level of excitement (passion, anger, perception of a danger etc.), orange with a lower level of excitement, yellow with joy, white with a neutral emotion, and gray with sadness (Fugate and Franco, 2019). These details have been added to the revised version.

Comment 9: In "Limitations" more shortcomings should be taken into account.

Response 9: Among the limitations, we cited the cross-sectional nature of the study, which does not allow causal links to be established, and the small sample size of volunteers. We added that it was a voluntary, non-randomized sample and that the male/female and normal weight/overweight/obese ratios were not balanced.

Reviewer 3 Report

Comments and Suggestions for Authors

The study by Rousset et al. aimed to examine the relationships between stress, behavior, and individual factors in middle-aged employees in the service sector. A strength of the study is the use of real-world data collected from phones and PCs. This enhances the ecological validity of the findings, increasing the real-world applicability of the results.

I have several comments and suggestions for further clarification and improvement:

  1. line 67, "This one-time study explored the individual characteristics." Does "one-time" refer to a cross-sectional study design? Please clarify this point.
  2. line 106, more details regarding the classification of physical activity intensity are encouraged. Specifically, please provide information on the identification of non-wear time, the accuracy of the algorithm used, and any relevant validation procedures.
  3. The authors identified dairy product consumption as a significant factor associated with home stress, which is interesting. In Figure 3, many factors are listed as potentially relevant. However, I feel that using stepwise regression may not be the most effective approach for identifying important factors related to stress. I suggest that the authors consider using principal component analysis (PCA) to better cluster and interpret these factors.

Author Response

Reviewer 3:

Comment 1: the study by Rousset et al. aimed to examine the relationships between stress, behavior, and individual factors in middle-aged employees in the service sector. A strength of the study is the use of real-world data collected from phones and PCs. This enhances the ecological validity of the findings, increasing the real-world applicability of the results.

Response 1: Thank you for your comments and compliments.

Comment 2: I have several comments and suggestions for further clarification and improvement:

  1. line 67, "This one-time study explored the individual characteristics." Does "one-time" refer to a cross-sectional study design? Please clarify this point.
  2. line 106, more details regarding the classification of physical activity intensity are encouraged. Specifically, please provide information on the identification of non-wear time, the accuracy of the algorithm used, and any relevant validation procedures.
  3. The authors identified dairy product consumption as a significant factor associated with home stress, which is interesting. In Figure 3, many factors are listed as potentially relevant. However, I feel that using stepwise regression may not be the most effective approach for identifying important factors related to stress. I suggest that the authors consider using principal component analysis (PCA) to better cluster and interpret these factors.

Response 2: This one-time study is a cross-sectional study design. We  have replaced “one time” by cross-sectional.

Using accelerometry data collected by smartphones, our algorithms are able to determine the amount of time spent immobile (sitting or standing), engaged in light activity (walking slowly), moderate activity (walking normally), or vigorous activity (running, sports activities). These algorithms have been validated in relation to energy expenditure and indirect calorimetry. The procedure for developing and validating these algorithms is explained in several publications, notably in Guidoux et al. (2014) below: The absolute errors in estimating the time spent in the four activity categories (sedentary, light-, moderate-, and vigorous-intensity activity) are less than 5% compared to indirect calorimetry in both normal-weight or overweight adults. Volunteers had to carry their phones while awake, i.e., between 8 a.m. and 10 p.m. They did not collect data during the night or while sleeping. Accelerometer data collection is automatic and continuous. It is performed by the smartphone as long as it is turned on.

                        Rousset S, Coyault Abele D, Benoit M, Zemni R, Lacomme P, Fleury G. (2020). Spontaneous Physical Activity and Sedentary Patterns Analyzed in a General Population of Adults by the eMouve Application. Springer Nature Switzerland AG 2020 T. Ahram et al. (Eds.) 1152, 363–368, doi: https://doi.org/10.1007/978-3-030-44267-5_54.

                        Paris L, Guidoux R, Saboul D, Duclos M, Boirie Y, Rousset S. (2019). Comparison of Active and Sedentary Bout Lengths in Normal and Overweight Adults using eMouveRecherche. International Journal of Sports and Exercice Medicine 5, (11), 151-160, doi: 10.23937/2469-5718/1510151.

                        Rousset S, Guidoux R, Paris L, Farigon N, Boirie Y, Lacomme P, Phan R, Ren L, Saboul D & Duclos M. (2018). eMouveRecherche: the first scientific application to promote light-intensity activity for the prevention of chronic diseases. Biology, Engineering and Medicine 3, (1), 1-6, doi: 10.15761/BEM.1000133.

                        Guidoux R, Duclos M, Fleury G, Lacomme P, Lamaudière N, Saboul D, Ren L & Rousset S. (2017). The eMouveRecherche application competes with research devices to evaluate energy expenditure, physical activity and still time in free-living conditions. Journal of Biomedical Informatics 69, 128-134, doi: 10.1016/j.jbi.2017.04.005.

                        Rousset S, Guidoux R, Paris L, Farigon N, Miolanne M, Lahaye C, Duclos M, Boirie Y & Saboul D. (2017). A novel smartphone accelerometer application for low-intensity activity and energy expenditure estimations in overweight and obese adults. Journal of Medical Systems 41, (117), 1-10, doi: 10.1007/s10916-017-0763-y.

                        Paris L, Duclos M, Guidoux R & Rousset S . (2016). Evaluation of physical activity and energy expenditure in overweight and obese adults. International Journal of Sports and Exercice Medicine 2, (3), 1-6.

                        Duclos M, Fleury G, Lacomme P, Phan R, Ren L, Rousset S . (2016). An acceleration vector variance based method for energy expenditure estimation in real-life environment with a Smartphone/Smartwatch integration. Expert Systems with Applications 63, 435-449, doi: 10.1016/j.eswa.2016.07.021.

                        Rousset S, Fardet A, Lacomme P, Normand S, Montaurier C, Boirie Y & Morio, B. (2015). Comparison of total energy expenditure assessed by two devices in controlled and free-living conditions. European Journal of Sport Science 15, (5), 391-399, doi: 10.1080/17461391.2014.949309.

                        Duclos M, Fleury G, Guidoux R, Lacomme P, Lamaudière N, Manenq P-H, Paris L, Ren L, Rousset S. (2015). Use of smartphone accelerometers and signal energy for estimating energy expenditure in daily-living conditions. Current Biotechnology 4, (1), 4-15, doi: http://www.eurekaselect.com/128913#.

                        Guidoux R, Duclos M, Fleury G, Lacomme P, Lamaudière N, Manenq P-H, Paris L, Ren L, Rousset S. (2014). A smartphone-driven methodology for estimating physical activities and energy expenditure in free living conditions. Journal of Biomedical Informatics 52, 271-278, doi: 10.1016/j.jbi.2014.07.009.

                        Guidoux R, Boualit R, Duclos M, Fleury G, Lacomme P, Lamaudière N, Rousset S. (2013). Conception d'une nouvelle fonction d'estimation de la dépense énergétique adaptée aux smartphones et aux conditions habituelles de vie. Nutrition clinique et métabolisme 27, S57.

We chose stepwise regression because it is a statistical decision-making analysis, whereas PCA is a qualitative analysis that provides descriptive information without indicating the significance of the variables.

Round 2

Reviewer 2 Report

Comments and Suggestions for Authors

Dear Authors, thank you for your answers and additions to the paper. Now it is much more comprehensive. I have only two more remarks:

1. Please include an information about the criteria  of obesity and overweight.

2. As the "Limitations", it's worth noting that data regarding weight and height are self-reported, not measured. Therefore, they may not be entirely reliable.

Author Response

Reviewer 2:

Comment 1: Please include an information about the criteria of obesity and overweight.

Response 1: Participants are considered overweight when their BMI is between 25 and 30 kg/m², and obese when their BMI is above 30. This sentence has been added in the revised manuscript.

Comment 2: As the "Limitations", it's worth noting that data regarding weight and height are self-reported, not measured. Therefore, they may not be entirely reliable.

Response 2: We agree with your comment. Weight is often underestimated when reported by the individual themselves. This limitation has been added in the revised manuscript.

Reviewer 3 Report

Comments and Suggestions for Authors

No further comments.

Author Response

No further comments.